# GaN-Based Resonant-Cavity Light-Emitting Diodes Grown on Si

**DOI:** 10.3390/nano12010134

**Published:** 2021-12-31

**Authors:** Wen Chen, Meixin Feng, Yongjun Tang, Jian Wang, Jianxun Liu, Qian Sun, Xumin Gao, Yongjin Wang, Hui Yang

**Affiliations:** 1Jiangxi Institute of Nanotechnology, Nanchang 330200, China; wenchen2020@sinano.ac.cn (W.C.); jxliu2018@sinano.ac.cn (J.L.); 2Key Laboratory of Nanodevices and Applications, Suzhou Institute of Nano-Tech and Nano-Bionics (SINANO), Chinese Academy of Sciences (CAS), Suzhou 215123, China; yjtang2018@sinano.ac.cn (Y.T.); jianwang2020@sinano.ac.cn (J.W.); hyang2006@sinano.ac.cn (H.Y.); 3Nano Science and Technology Institute, University of Science and Technology of China, Suzhou 215123, China; 4Guangdong Institute of Semiconductor Micro-Nano Manufacturing Technology, Foshan 528000, China; 5Grünberg Research Centre, Nanjing University of Posts and Telecommunications, Nanjing 210003, China; gaoxm@njupt.edu.cn (X.G.); wangyj@njupt.edu.cn (Y.W.)

**Keywords:** GaN-on-Si, resonant-cavity light-emitting diode, chemical-mechanical polishing, visible light communications

## Abstract

GaN-on-Si resonant-cavity light-emitting diodes (RCLEDs) have been successfully fabricated through wafer bonding and Si substrate removal. By combining the chemical mechanical polishing technique, we obtained a roughness of about 0.24 nm for a scan area of 5 μm × 5 μm. The double-sided dielectric distributed Bragg reflectors could form a high-quality optical resonant cavity, and the cavity modes exhibited a linewidth of 1 nm at the peak wavelength of around 405 nm, corresponding to a quality factor of 405. High data transmission in free space with an opening in the eye diagram was exhibited at 150 Mbps, which is limited by the detection system. These results showed that GaN-based RCLEDs grown on Si are promising as a low-cost emitter for visible light communications in future.

## 1. Introduction

GaN-based resonant-cavity light-emitting diodes (RCLEDs) with narrow spectral width, stable peak wavelength, superior directionality, and high output-coupling efficiency [1,2,3,4] have shown great potential applications in various fields, such as line-of-sight visible light communications (VLCs), plastic optical fiber-based networks, optical scanners, speckle-free illumination, printers, and displays [5,6,7,8].

Up to now, there have been no GaN-based RCLED products. The main bottleneck is that most GaN-based RCLEDs are grown on sapphire substrates, resulting in several issues. For GaN-based RCLEDs with hybrid distributed Bragg reflectors (DBRs), about 40 pairs of GaN/AlGaN or GaN/InAlN DBRs are usually grown underneath the quantum wells (QWs) [9,10,11,12], which may not only induce high tensile stress and even micro-cracks, but also increase the dislocation density, seriously reducing the internal quantum efficiency [13]. For GaN-based RCLEDs with double-sided dielectric DBRs, the laser lift-off process is usually used to remove the sapphire substrates [14,15], which often causes high stress and even cracks, greatly affecting the device performance and yield. Compared with sapphire substrates, Si substrates can be easily removed by wet chemical etching without any damage. Si substrates also have unique advantages in wafer size and material cost, as well as the use of depreciated automation processing lines [16]. Lastly, Si substrates match with the conductive Si submount in the coefficient of thermal expansion, which can greatly reduce the stress in the bonding process and hence improve the device performance and yield. Therefore, replacing the small-sized (≤6 inches) sapphire substrates with large-diameter (up to 20 inches) cost-effective Si substrates is expected to greatly slash the device fabrication cost, and increase the device performance and yield, ultimately realizing the mass production of GaN-based RCLEDs.

For this paper, we fabricated GaN-based RCLEDs grown on Si substrate through wafer bonding and Si substrate removal. In order to improve the quality of the optical resonant cavity and reduce the optical loss, a chemical mechanical polishing (CMP) process was developed to reduce the roughness of the N-face n-GaN surface for the n-side DBR deposition. With these technologies, a high-quality optical resonant cavity could be formed by double-sided DBRs. Afterwards, the optical and electrical performance of the as-fabricated GaN-based RCLED was characterized.

## 2. Materials and Methods

The GaN-based RCLED material was epitaxially grown on Si(111) substrate by metal-organic chemical vapor deposition. A 1-μm-thick Al-composition graded AlN/AlGaN multilayer buffer was firstly grown to establish the compression strain to not only compensate for the tensile stress due to the difference in thermal expansion coefficient, but also to effectively filter out the threading dislocations induced by the lattice mismatch between GaN and Si [17,18,19]. By using this technology, a crack-free high-quality GaN-on-Si template was grown. Upon this template, the RCLED epitaxial structure was grown, including a 1.3-μm-thick n-type GaN contact layer, a 1.3-μm-thick n-type AlGaN layer, an 80-nm-thick n-type GaN waveguide, three pairs of InGaN (2.5 nm)/GaN (10 nm) QWs, a 60-nm-thick undoped GaN waveguide, a 20-nm-thick p-type AlGaN electron blocking layer, a 600-nm-thick p-type AlGaN layer, and a 30-nm-thick p-type heavily doped GaN contact layer. The RCLED device was processed in a vertical structure, and the p- and n-contact pads were distributed at the two opposite sides, as shown in Figure 1.

Figure 2 shows the detailed fabrication process flow of the GaN-based RCLEDs. The process included the following: (1) dry etching of the p^++^-GaN layers to form a mesa with a diameter of 20 μm and a depth of 30 nm by using Cl-based inductively coupled plasma (ICP) etching; (2) depositing a 30-nm-thick SiO_2_ layer by using ICP chemical vapor deposition as a current confinement layer to form the current aperture region; (3) evaporating a 30-nm-thick indium tin oxide (ITO) film to form good ohmic contact with the p^++^-GaN and also work as the current spreading layer; (4) depositing a 12.5-pair TiO_2_/SiO_2_ multilayer dielectric DBR on the ITO film; (5) bonding the as-fabricated GaN-on-Si RCLED wafer to a conductive Si wafer with a AuSn solder; (6) utilizing wet chemical etching and ICP dry etching to remove the Si(111) substrate and AlN/AlGaN multilayer buffer; (7) developing a CMP technique to smooth the rough N-face n-GaN surface with the silica solution in order to reduce the scattering loss of the optical cavity; (8) ICP etching to form the N-face n-type GaN mesa; (9) using fluorine ion implantation to implement the insulation between the RCLED devices; (10) depositing a 11.5-pair TiO_2_/SiO_2_ patterned multilayer dielectric DBR on the n-GaN surface, followed by the patterned Ti/Pt/Au metal stack with thicknesses of 50/50/100 nm as the n-type ohmic contact electrode.

The surface roughness of these samples was characterized by atomic force microscopy (Veeco, Plainview, NY, USA). The current–voltage characteristics of the RCLEDs were measured using the Keithley 4200-SCS/F parameter analyzer. The electroluminescence (EL) spectra and output power were collected by a spectrometer (FX 4000, Ideaoptics, Shanghai, China) and a Si photodetector, respectively. The emission images of the RCLEDs were obtained by optical microscope (Optiphot 200, Nikon, Tokyo, Japan).The scanning electron microscopy (SEM) image was obtained by cold-field emission scanning electron microscope (S-4800, HITACHI, Tokyo, Japan). The cross-sectional scanning transmission electron microscopy (STEM) image was obtained by focused ion beam (FEI Scios, FEI, Hillsboro, OR, USA) and field emission transmission electron microscope (FEI Talos F200X, FEI, Hillsboro, OR, USA). The high-speed data transmission feasibility was examined using an arbitrary waveform generator (Agilent 33522A, Agilent, Santa Clara, CA, USA) to generate a pseudo-random binary sequence (PRBS) data stream, and the PRBS was processed in a bias-tee circuit and fed to the RCLED. The light signals were converted into electrical signals by a photodiode module (Hamamatsu C12702-11, Hamamatsu Inc., Hamamatsu, Japan), and the amplified electrical signals were displayed by a digital storage oscilloscope (Agilent DSO9254A, Agilent, Santa Clara, CA, USA).

## 3. Results and Discussion

A rough surface will cause a large optical scattering loss, and hence ultimately affect the quality of the resonant cavity. Therefore, it is necessary to obtain a flat morphology to deposit dielectric DBRs. However, the Cl-based plasmas in the ICP etching often cause an uneven etching, especially for high-Al-composition AlGaN materials, which increases the surface roughness [20]. After removing the AlN/AlGaN multilayer buffer by ICP etching, the surface roughness of the N-face n-GaN was about 3.56 nm for a scan area of 5 μm × 5 μm, as shown in Figure 3a. To smooth the surface morphology, a CMP technology with silica solution was developed. The alkaline silica solution could preliminarily corrode and soften the N-face n-GaN surface, and then the silica nanoparticles in the solution could polish and smooth the N-face n-GaN surface [21]. Figure 3b shows the surface morphology of N-face n-GaN after polishing 1200 nm with the silica solution for 30 min. The roughness of a scan area of 5 μm × 5 μm was greatly reduced from 3.56 to 0.24 nm, which was further evidenced by the cross-sectional STEM image of the as-fabricated device as shown in Figure 3c. This surface smoothness was comparable to that of the as-grown Ga-face GaN, and paved the way for the deposition of high-quality n-side dielectric DBRs [22]. As a result, the 11.5-pair TiO_2_/SiO_2_ n-DBRs showed a peak reflectivity of 99% at 405 nm with a large stop-band of about 80 nm, as shown in Figure 3d, which is almost equal to that of the 12.5-pair p-DBRs (99.5%).

Figure 4a presents the measured light–current–voltage (L–I–V) plots. The I-V curve of the as-fabricated RCLED exhibited a turn-on voltage of approximately 3 V. Above the turn-on voltage, the current density gradually increased with voltage. The inset of Figure 4a presents the leakage current under a reverse bias. At −5 V, the reverse leakage current was as low as 0.34 nA, which indicates that the fluoride ion implantation effectively insulated the devices. The fluoride ion with a very strong electronegativity could not only trap free electrons, but also reduce the carrier density by host-type defects at ultra-deep energy levels, ultimately leading to the insulation of the devices [23]. Figure 4b shows the EL spectrum of the as-fabricated RCLED at an injection current of 7 mA. Compared with conventional LEDs, the FWHM of the EL spectrum was reduced from about 15.5 to 10.6 nm due to the resonant cavity effect [24]. On the other hand, as shown in Figure 4b, the EL spectrum of the RCLED was periodically modulated by the Fabry–Pérot (F–P) cavity formed by the double-sided dielectric DBRs [25], and showed several narrow resonant peaks with a spacing of about 4 nm. This was mainly due to the interference effect as light bounced back and forth between the double-sided DBRs until the constructive conditions achieved allowed the cavity modes to escape from the LEDs [26]. Based on the analysis method described in [7], the inset of Figure 4b shows a dominant resonant peak at a wavelength of about 405 nm with a FWHM of approximately 1 nm, corresponding to a quality factor of 405. For the as-fabricated GaN-based RCLED, with a mode spacing of 4 nm, the cavity length of the RCLED was calculated as 5.05 μm at the peak wavelength of λ_0_ = 405 nm [27,28]. This value was larger than the total thickness of the GaN-based epilayers (4.2 μm) measured by STEM, which was mainly because the effective resonant cavity was composed of not only the remaining GaN-based epilayers but also the double-sided DBR mirrors. For the as-fabricated GaN-based RCLED, part of the optical field would penetrate into the double-sided dielectric DBRs, which contributed to the increased effective cavity length [26]. Figure 4c presents the peak wavelength of the device under various injection currents. For the GaN-based emitter with similar epitaxial structure [29], the peak wavelength of the EL spectrum was blue-shifted from 416.2 to 412.7 nm as the injection current density was increased from 1.5 to 4.7 kA/cm^2^, which was mainly due to the screened quantum-confined Stark effect. In contrast, for the as-fabricated GaN-based RCLED, the peak wavelength only decreased from 405.2 to 404.7 nm while the injection current density increased from 1.45 to 4.36 kA/cm^2^. Thanks to the short cavity formed by the double-sided DBRs, the GaN-based RCLED showed a much more stable peak wavelength.

Figure 5a,b shows the top-view SEM and the light-up images of the as-fabricated GaN-based RCLED, respectively. It can be clearly seen that the circular shape of the 20-μm-diameter device was well preserved, and uniform luminescence was achieved by using a 30-nm-thick ITO current spreading layer. In Figure 5b, the peripheral blue luminescence was mainly due to the light scattering by the peripheral of the n-side DBRs.

The high-speed data transmission feasibility of the as-fabricated GaN-based RCLED was examined by a free space visible light communication (VLC) measurement system. Figure 6a shows the open eye diagram of the as-fabricated GaN-based RCLED measured at 150 Mbps, and Figure 6b shows the waveform diagram of optical signal and electrical signal. It can be clearly seen that the optical signal matches with the electrical signal very well, showing a good transmission feasibility. It should be clearly noted that the transmission rate of 150 Mbps was the maximum value of the detection system, and we believe that the actual maximum transmission rate could be even higher.

## 4. Conclusions

In summary, we have grown GaN-based RCLED materials on Si substrate and then fabricated the GaN-on-Si RCLEDs through wafer bonding and Si substrate removal. By adopting a CMP technique, the surface roughness of the N-face n-GaN was greatly reduced to 0.24 nm for a scan area of 5 μm × 5 μm, which laid down a great foundation for the deposition of dielectric DBRs. The flat surface together with the double-sided dielectric DBRs formed a high-quality optical resonator, which narrowed and periodically modulated the EL spectra, while stabilizing the peak wavelength. In addition, the as-fabricated GaN-based RCLED experimentally showed a PRBS transmission rate of 150 Mbps, which was limited by the detecting system and could be even higher. These experimental results show that GaN-based RCLEDs grown on Si are promising as a low-cost emitter for POF communication, solid-state lighting, and VLC applications in future.

## Figures and Tables

**Figure 1 nanomaterials-12-00134-f001:**
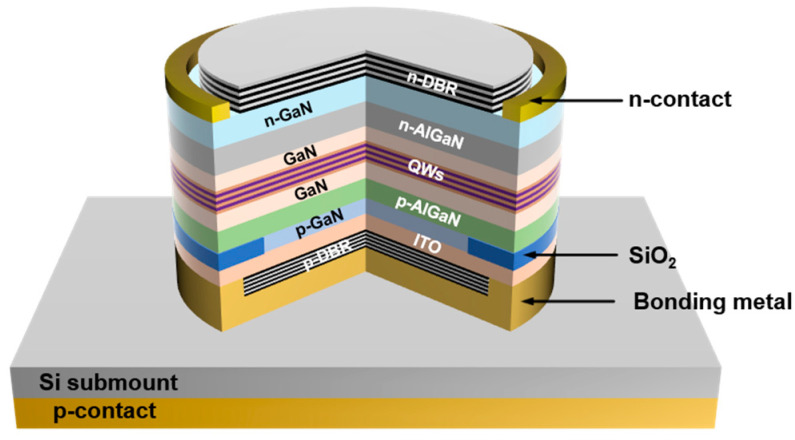
Schematic diagram of the GaN-based RCLED.

**Figure 2 nanomaterials-12-00134-f002:**
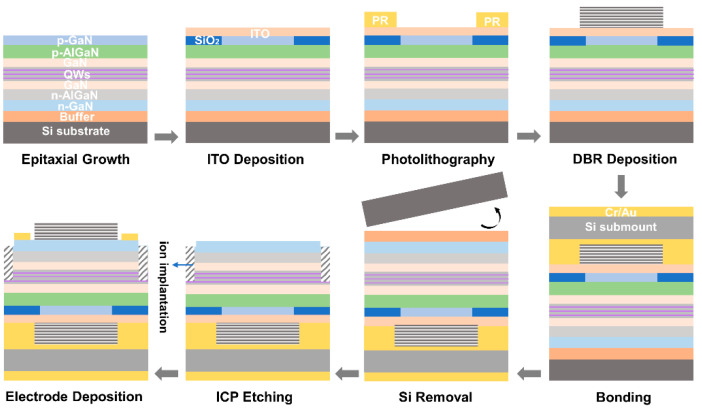
The detailed fabrication process flow of the GaN-based RCLED grown on Si.

**Figure 3 nanomaterials-12-00134-f003:**
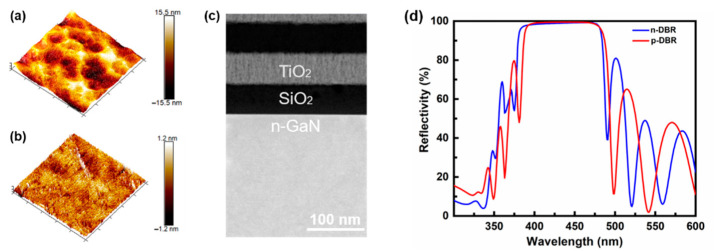
AFM images of the N-face n-GaN surface (**a**) before and (**b**) after the CMP with silica solution showing a scan area of 5 × 5 μm^2^. (**c**) Cross-sectional STEM image of the interface between the N-face n-GaN and the n-side DBR. (**d**) Experimental reflectance spectra of the n-DBR and p-DBR.

**Figure 4 nanomaterials-12-00134-f004:**
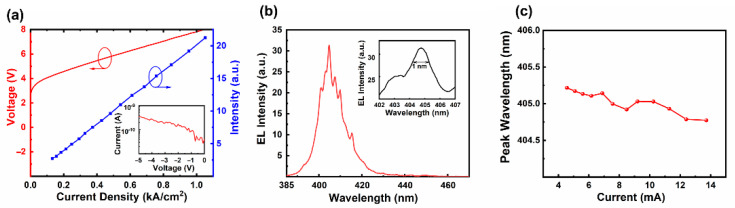
(**a**) Measured L–I–V characteristics of the as-fabricated GaN-based RCLED, and the inset showing leakage current under a reverse bias. (**b**) EL spectrum of the RCLED at an injection current of 7 mA, and the inset showing the dominant cavity mode with a linewidth of approximately 1 nm. The silicon detector was located 1 cm directly above the device and collected only a portion of the output power. (**c**) The peak wavelength of RCLED at various injection currents.

**Figure 5 nanomaterials-12-00134-f005:**
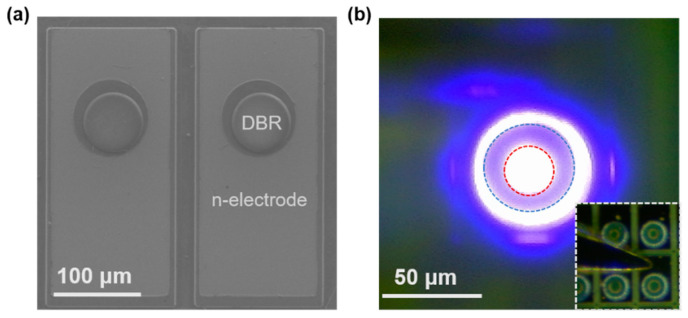
(**a**) Top-view SEM and (**b**) light-up images of the as-fabricated GaN-based RCLED with a diameter of 20 μm. In (**b**), the injection current was 2 mA, and the red and blue circles show the sizes of the current aperture and the n-side DBR, respectively. The inset shows the optical microscopy image of the RCLED.

**Figure 6 nanomaterials-12-00134-f006:**
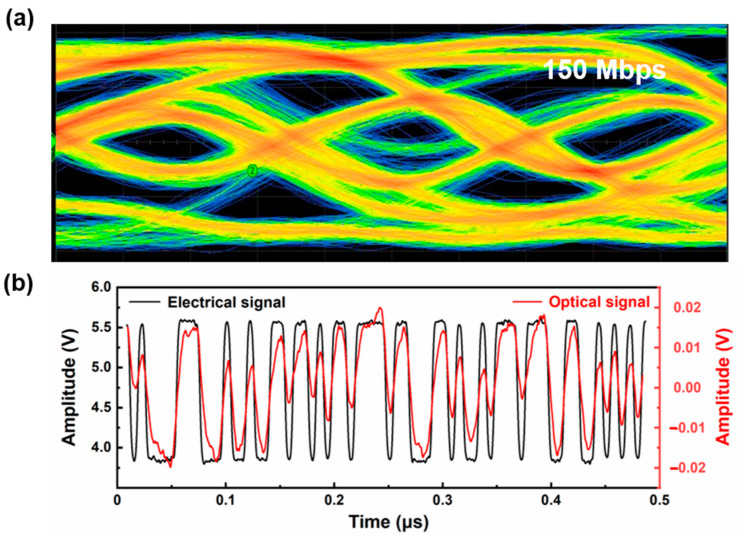
(**a**) Measured eye diagram of the as-fabricated GaN-based RCLED at a transmission rate of 150 Mbps. (**b**) The waveform diagram of optical signal and electrical signal at a transmission rate of 150 Mbps.

## Data Availability

The data presented in this study are available on request from the corresponding author.

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
