# Peer review of "GaN-Based Resonant-Cavity Light-Emitting Diodes Grown on Si"

_nanomaterials, 2021, doi:10.3390/nano12010134_

Round 1

Reviewer 1 Report

The topic is timely and relevant. The research is well described. Two issues ought to be clarified.

First - the depth of mesa formed on p-type side of GaN-based  RCLED should be given precisely. On line 73 we find "dry etching of p++-GaN and p-AlGaN layers to form a mesa ....." while in Figs 1 and  2 it looks like a mesa is etched in n p++GaN only.

Second - Fig. 4 a and Fig. 4b are far too small. In particular the shape of the spectrum is hardly visible.

Reviewer 2 Report

Reviewer report:

Wen Chen et al reported the manuscript entitled “GaN-based Resonant-Cavity Light-Emitting Diodes Grown on Si.” The manuscript presents the fabrication and characterization of GaN-on-Si resonant-cavity light-emitting diodes (RCLEDs). The RCLEDS contains double-sided dielectric distributed Bragg reflectors forming an optical resonant cavity, and the cavity modes exhibited a linewidth of 1 nm at the peak wavelength of around 405 nm, corresponding to a quality factor of 405. The author claims that the GaN-based RCLEDs showed a high data transmission in free space with an opening in the eye diagram at 150 Mbps. The authors should clearly mention the novelty of the results by comparing them with previously reported similar works. The significance of the work is not very high enough. This work is only the optimization of the methodology, and the OLED performance is not exciting, which will abate the novelty of the manuscript. The authors further need to improve the manuscript before submitting it by considering the following points. The reviewer recommends a minor revision to straighten the quality of the study.

  1. What is the purpose of using the DBR resonant cavity in this study? The EL spectrum is broad and the 1 nm narrowing is just a feature in the EL. It is not clear what is the role of the cavity.
  2. The author claims that the quality of the optical resonant cavity is improved and the optical loss is reduced. What is the evidence of this claim? Could you compare the quality factor with state-of-the-art to comfort this claim?
  3. The originality is lacking in this paper. There are no clear evidence and physical explanation.
